# Trophic Position of the Species and Site Trophic State Affect Diet Niche and Individual Specialization: From Apex Predator to Herbivore

**DOI:** 10.3390/biology12081113

**Published:** 2023-08-10

**Authors:** Lukáš Vejřík, Ivana Vejříková, Petr Blabolil, Zuzana Sajdlová, Luboš Kočvara, Tomáš Kolařík, Daniel Bartoň, Tomáš Jůza, Marek Šmejkal, Jiří Peterka, Martin Čech

**Affiliations:** 1Biology Centre of the Czech Academy of Sciences, Institute of Hydrobiology, Na Sádkách 7, 37005 České Budějovice, Czech Republic; lukas.vejrik@hbu.cas.cz (L.V.); petr.blabolil@hbu.cas.cz (P.B.); zuzana.sajdlova@hbu.cas.cz (Z.S.); lubos.kocvara@hbu.cas.cz (L.K.); kolarik.tom1@seznam.cz (T.K.); daniel.barton@hbu.cas.cz (D.B.); tomas.juza@hbu.cas.cz (T.J.); marek.smejkal@hbu.cas.cz (M.Š.); jiri.peterka@hbu.cas.cz (J.P.); martin.cech@hbu.cas.cz (M.Č.); 2Faculty of Science, University of South Bohemia in České Budějovice, Branišovská 31, 37005 České Budějovice, Czech Republic

**Keywords:** aquatic food web, freshwater ecosystem, isotopic half-life, niche width, stable isotope analysis

## Abstract

**Simple Summary:**

Niche widths and individual specialization were studied based on the isotopic signals, but using the innovative and non-lethal approach. We analyzed four different body tissues with different isotopic half-lives, and revealed crucial results on trophic interactions of fish. We assume that the observed trends will occur in other food webs with similar trophic positions. For example, the apex predator status of ectotherms is linked to the individual size, not to the species. In addition, thanks to the consideration of the site trophic state (total phosphorus content), which has been little studied in relation to niche width, we observed a significant impact on the individual specialization of species in higher trophic positions. Thus, eutrophication can significantly change the foraging behavior.

**Abstract:**

Intra-species variability in isotopic niches, specifically isotopic total niche width (ITNW), isotopic individual niche width (IINW), and isotopic individual specialization (IIS), was studied using an innovative approach without sacrificing the vertebrates. Stable isotopes (*δ*^13^C, *δ*^15^N) in four body tissues differing in isotopic half-life were analyzed from four freshwater fish species representing different trophic positions. ITNW was widest for the apex predator (European catfish) and narrowest for the obligate predator (Northern pike). IINW exhibited a polynomial trend for the European catfish, Northern pike, and Eurasian perch (mesopredator), decreasing with body mass and increasing again after exceeding a certain species-dependent body mass threshold. Thus, for ectotherms, apex predator status is linked rather to its size than to the species. In herbivores (rudd), IINW increased with body mass. The IIS of predators negatively correlated with site trophic state. Therefore, eutrophication can significantly change the foraging behavior of certain species. We assume that the observed trends will occur in other species at similar trophic positions in either aquatic or terrestrial systems. For confirmation, we recommend conducting a similar study on other species in different habitats.

## 1. Introduction

Food selection is a critical component of life at the individual, population, and species level, and is one of the crucial aspects defining the ecological niche of an organism [1,2]. The discovery of niche variations between individuals within populations meant a significant breakthrough in the understanding of the food ecology of species [3]. Populations of ‘generalist’ species may in fact be a collection of individual-level trophic specialists that vary considerably in their resource use [4,5]. This applies even to humans, the genus Homo was assigned a ‘generalist specialist’ niche, defined as the ability to both generalize as a species and specialize as individuals in one’s environment [6].

In the case of water animals, for example, Eurasian perch (*Perca fluviatilis*), which have a wide niche width at the population level, segregate into littoral and pelagic specialists, and consequently individuals do not couple these two components of freshwater food webs [7]. At the level of apex predators (bull sharks, *Carcharhinus leucas*, and tiger sharks, *Galeocerdo cuvier*), despite wide population-level isotopic niche breadths in both species, isotopic values of individual tiger sharks varied across tissues with different isotopic half-life. The population niche breadth was explained mostly by variation within individuals, suggesting tiger sharks are true generalists. In contrast, isotope values of individual bull sharks were stable through time and their wide population level niche breadth was explained by variation among specialist individuals [8]. Similarly, according to the isotope values of individual large freshwater predators (European catfish *Silurus glanis*, Northern pike *Esox lucius*), European catfish seem to be true generalists, similar to tiger sharks, whereas a high degree of individualism and also population specialization was revealed for Northern pike [9]. In contrast, the food specialization of rudd (*Scardinius erythrophthalmus*) is significantly dependent on temperature. Rudd is almost a strict herbivore in the warm and becomes a benthivore or an occasional predator during the cold period of the year [10,11].

In general, it seems that predators expected to have high requirements for learning specific hunting skills become stricter food specialists, both as species and individuals [12]. There is a strong trade-off between specializing on one versus many prey species, with selection acting against individual generalists [12,13]. As mentioned by DeSantis et al. [2], pursuing voles, rabbits and deer may make an individual predator a ‘jack of all trades and master of none’. Herbivorous generalists have received less attention than predators [12]. It was assumed that grazers are not pushed to become foraging specialists due to sufficient food availability [14]. Nevertheless, a recent study demonstrated that individuals among grazers are specialized and less representative of their overall species’ dietary breadth [2]. Individual specialization probably reduces intraspecific competition, increases carrying capacities, and may have stabilizing effects on species and communities over time [2].

The total niche width (TNW) of a species (or population) depends strongly on the variation among competing individuals [15]. According to Araújo et al. [16], co-occurring individuals may utilize only a part of TNW for several reasons, such as different physiological requirements [17], different optimization criteria (prioritizing maximum energy intake while ignoring high predation risk or vice versa; [18]), different phenotypic variation related to ontogeny (thus different ability to detect, capture, handle, and digest alternative prey), or different ability to access food sources due to social dominance [19].

Individual niche width (INW) can be defined by Optimal Foraging Theory [20]. Each individual inhabits an environment with potential food sources that set an upper limit on INW, and the theory states that an individual will use the most profitable food sources. When a preferred food source becomes scarce, the individual begins to use previously ignored sources. An active selection of a small subset of TNW from a shared environment can be described as individual specialization (IS) [21]. According to the review [16], INWs are found to match the TNWs by only 47% on average. The degree of IS is presumed to be affected by intraspecific competition, which tends to increase IS [22,23]; interspecific competition, on the contrary, likely weakens IS [24]. Last but not least, when resources are limited and the trophic state of environment is low, individuals are forced to specialize more [16,25].

Studies dealing with niche width and IS usually compare two species in a similar trophic position [8,9,26]. However, no studies have focused on TNW, INW, and IS across different trophic positions from herbivore to apex predator. Therefore, we have chosen four fish species ranging from the apex predator European catfish, obligate predator Northern pike [9], and the mesopredator Eurasian perch [27], to mostly herbivorous rudd, the species with the lowest trophic position [11]. Fish were selected as the model organisms, because they are the most commonly used group in stable isotope-based food web studies [16]. The study considers site trophic state (total phosphorus content, TP, a basic and key indicator of water trophic state; [28,29]), which has been little studied in relation to niche width and IS [30].

All studied metrics, i.e., TNW, INW, and IS, are commonly based on the food intake of individuals over time [16]. However, for instance, Sheppard et al. [31] determined IS based on the stable isotope analysis (SIA) of *δ*^13^C and *δ*^15^N. In the aquatic environment, highly cited Matich et al. [8] focused on the determination of TNW, INW, and IS metrics in marine apex predators based on the *δ*^13^C and *δ*^15^N of three different body tissues differing in isotopic half-life. Although this study is truly innovative and highly respected, it contained several methodological limitations, such as only two studied species, which were studied in geographically different locations. Another limitation was the use of different body tissues without taking into account the discrimination factor of the different tissues to have comparable *δ*^13^C and *δ*^15^N [32]. Further, the studies [8,31] used SIA equivalently to the diet analysis, but the diet and isotopic niches are not identical [33]. As the use of stable isotopes in connection with the individual specialization has recently been increasing, our study is focused on the isotopic individual specialization and tries to eliminate the so far revealed deficiencies related to the SIA. Thus, the study represents another methodological step in the analysis of isotopic total niche width (ITNW), isotopic individual niche width (IINW), and isotopic individual specialization (IIS).

We aimed to test the following hypotheses: (i) the ITNW of a species and a population will increase with both trophic position in the food web and with the site trophic state. (ii) IINW will decrease with the body mass increase in individuals as they become more specialized. (iii) IIS will increase gradually with trophic position, but decrease rapidly at the apex predator position, where a high generalism is expected (apex predator effect). (iv) IIS will exhibit short-term (seasonal) fluctuations with a decreasing trend in the degree of IIS over time. As the availability of food sources changes, individuals switch to other more readily available sources. For this reason, the IIS will be lower when comparing tissues with large differences in isotopic half-life.

## 2. Materials and Methods

### 2.1. Study Design

The study was conducted on four waterbodies with comparable size and species composition but differing in trophic state: two oligotrophic lakes, Most and Milada, and two mesotrophic reservoirs, Žlutice and Římov, Czech Republic (Graphical Abstract). See Table 1 for basic parameters, and Vejřík et al. [34] for more details about the sites. Total phosphorus (TP) presents a long-term average from longitudinal profiles of three areas sampled four times per year in 2013–2017. The sites were numbered 1–4 (i.e., Most, Milada, Žlutice, and Římov) according to their trophic state.

In 2017 electrofishing [35] was used with intention to capture 15 adult (i.e., sexually mature) individuals of four fish species in the period 4–7 September in the Most lake, 11–14 September in the Milada lake, 25–28 July in the Žlutice reservoir, and 18–21 July in the Římov reservoir. Fieldwork was timed right after the peak of seasonal feeding activity in the temperate climate zone.

In several cases, it was not possible to catch 15 individuals, but the catch was never fewer than 11 individuals. The body mass of the European catfish ranged from 1.6 to 21.8 kg (mean ± SD: 8.35 ± 5.18 kg), Northern pike 0.6–13.8 kg (3.75 ± 3.06 kg), Eurasian perch 0.15–1.9 kg (0.39 ± 0.25 kg), and rudd 0.16–1.29 kg (0.44 ± 0.25 kg) (Table 2). All individuals were measured to the nearest 0.5 cm. Smaller individuals up to 3 kg were weighed on ‘TSCALE QHW scales, 15 kg’ to the nearest gram and all other individuals were weighed in a pan on a platform scale ‘SOEHNLE Professional 6858, 200 kg’ to the nearest 50 g. Samples of four body tissues were collected under anesthesia for SIA. Specifically, a small sample of the anal fin (1 cm^2^) was collected (resection), as well as a 0.2 cm^3^ of dorsal muscle (biopsy punch, skin removed). Using an 18-gauge needle, 3 mL of blood was collected from the caudal vein. Next, 2 mL of blood was placed into BD Vacutainer blood collection vials and centrifuged at 3000 rpm for 1 min, and the separated plasma was collected. The remaining 1 mL of blood was preserved in its original composition for subsequent analysis. The four tissue samples from each individual were placed on ice and transferred to the laboratory freezer for SIA. No fish were sacrificed during the study and all were released back into the water bodies. The trophic positions of the studied species (not the same individuals) were determined based on both gut content and SIA published in previous studies from the same study sites [9,36]: European catfish as an apex predator, Northern pike as an obligate predator, Eurasian perch as a mesopredator, and rudd as the species with the lowest trophic position, mostly herbivorous [9,36].

Primary consumers, such as filter-feeding zebra mussels (*Dreissena polymorpha*), aquatic snails (*Radix auricularia*, *Planobarius* sp.), and water lice (*Asellus aquaticus*) were collected using tweezers, about 10–15 individuals depending on their size. Shells of the mollusks were removed. The samples were placed in Ziplock bags and frozen.

### 2.2. Stable Isotope Analysis

All frozen samples for SIA were later dried at 60 °C for 48 h and ground into a homogenous powder using a Retsch MM 200 ball-mill (Retsch GmbH, Haan, Germany). Small subsamples (0.52–0.77 mg) were weighed in tin cups for *δ*^13^C and *δ*^15^N analysis. All SIA were conducted using a FlashEA 1112 elemental analyzer coupled to a Thermo Finnigan DELTAplus Advantage mass spectrometer (Thermo Fisher Scientific Corporation, Waltham, MA, USA) at the University of Jyväskylä, Finland. Carbon and nitrogen isotope ratios are expressed as *δ*^13^C and *δ*^15^N relative to the international standards for carbon (Vienna PeeDeeBelemnite, Vienna, Austria) and nitrogen (atmospheric nitrogen). Analytical precision was ±0.20‰, determined by repeated analysis of a working standard (Northern pike white muscle tissue) inserted in each run after every five samples. C:N ratios were consistently lower than 3.5 (i.e., >90% of cases).

### 2.3. Data Processing and Statistics

#### 2.3.1. Data Centering

We are introducing a novel and more reliable method of determining the diet–tissue discrimination factor by centering individual body tissues (plasma, blood, fin) on the key tissue, the muscle (Figure 1). Although the discrimination factor was not taken into account in the prior study focused on IS based on different body tissues [8], we believe it is a necessary inclusion. It means that if the entire population utilizes the same type of prey, the isotopic signal from one type of tissue, for example, muscle, will be identical for all individuals. However, the isotopic signal from another tissue, for example, blood, will again be identical for all individuals, but different from the muscle isotopic signal. The signals from muscle and blood will always differ from each other even within one individual due to the different protein composition of the tissues. Different proteins have different diet–tissue discrimination factors for both *δ*^13^C and *δ*^15^N [32]. Centering addresses the issue of natural tissue differences in each species and allows the final values to reflect the real differences in the individual’s food intake variations in time and between sites more accurately. The muscle was chosen as the key tissue because it is generally the recommended tissue in SIA [37]. It would be actually possible to choose any tissue other than muscle as the key tissue for centering. The resulting difference in both *δ*^13^C and *δ*^15^N values between the tissues within individual and between individuals would always remain the same. The relative position of the individual samples to each other would therefore not change. Only the absolute position of all tissues would differ, which is irrelevant in the calculations.

A clear trend in the discrimination of individual body tissues was observed in the values of *δ*^13^C and *δ*^15^N, consistent for each species. Thus, the difference between the diameters of two central quartiles (values of 7 individuals, that is 46.7% of the total that were the closest to the mean value of *δ*^13^C and *δ*^15^N from all 15 individuals) between the tissue (plasma, blood, fin) and the muscle tissue was subtracted or added. The use of only 7 values, the closest to the mean value, from all 15 samples was chosen in order to minimize the possibility of false tissue discrimination caused by marginal outliers that may be affected by an intensive short-term switch in diet at a certain time. Centering was provided for both *δ*^13^C and *δ*^15^N, and for each lake separately to account for the different between-tissue variations among sites [38]. See Figure 1 for more details.

#### 2.3.2. Isotopic Baseline Corrections and Subsequent Determination of ITNW and IINW

Since *δ*^15^N and *δ*^13^C for basal resources can vary considerably among sites (e.g., [39]), the results from SIA were corrected for this variability prior to statistical analysis. The trophic position of fish was calculated according to Anderson and Cabana [40]:$${\mathrm{TP}}_{f}=\frac{\delta^{15}N_{f}-\delta^{15}N_{\mathrm{baseline}}}{3.4}+2$$
where TP_f_ means trophic position of fish, *δ*^15^N_f_ is the nitrogen isotopic ratio of fish, *δ*^15^N_baseline_ is the nitrogen isotopic ratio of primary consumers, 3.4 is one trophic level increment in *δ*^15^N, and 2 is the trophic position of the organism used to estimate the baseline (i.e., the primary consumer). We used the mean *δ*^15^N of several primary consumer individuals, such as filter-feeding zebra mussels (*Dreissena polymorpha*), aquatic snails (*Radix auricularia*, *Planobarius* sp.), and water lice (*Asellus aquaticus*), in line with previous research in this field (e.g., [41,42]). Although herbivory prevails during the year, *δ*^15^N values were clearly above the primary consumers at all study sites, and the three other fish species were always above the rudd.

We did not use the *δ*^13^C of basal resources (e.g., detritus or periphyton) because it can vary between sites and influence consumer signals without reflecting any real difference in consumer diet. Thus, we corrected for differences in basal resources for carbon (*δ*^13^Ccorr) according to Olsson et al. [43]:$$\delta^{13}C_{\mathrm{corr}}=\frac{\delta^{13}C_{f}-\delta^{13}C_{\mathrm{meaninv}}}{{\mathrm{CR}}_{\mathrm{inv}}}$$
where *δ*^13^C_f_ is the carbon isotope signal of fish, *δ*^13^C_meaninv_ is the mean invertebrate (from the surber samples) carbon isotope signal, and CR_inv_ is the carbon range (*δ*^13^Cmax–*δ*^13^Cmin) for the same primary consumers used for the baseline in calculating TP_f_.

ITNW and IINW were calculated from the corrected carbon and nitrogen stable isotope signals by calculating the total convex hull area encompassed by the smallest polygon containing the individuals in a population in the corrected *δ*^13^C and *δ*^15^N niche space [44] (Figure 2). The SIBER package (Stable Isotope Bayesian Ellipses in R; version 2.1.6. [45]) was used to estimate sample-size corrected standard ellipse areas (SEAcs), total convex hull areas (TAs), and proportional overlap of the SEAc areas. SEAcs from the isotopic signals of all individuals of one species determined the ITNW of the species at the site. SEAcs from isotopic signals of all four body tissues from one individual determined the IINW of the individual.

#### 2.3.3. Degree of Individual Specialization (IIS)

Specialization is calculated as the dietary variation within individuals (IINW) and between individuals (IBIC: isotopic between individual component of variation) of a population. The IINW of a population shows how variable an individual’s diet is over time based on the different isotopic half-life of the four body tissues (Figure 2). It is typically expressed as a mean value for the entire population but can be similarly assessed for individuals. The IBIC of a population shows how each individual’s diet differs from other members of the population [46] and is calculated using data obtained from four tissue samples of mostly 15 individuals (fewer in the following cases due to a limited catch: 12 Northern pike at site 1; 14 Northern pike at site 2; 11 Eurasian perch at site 4; 11 rudd at site 3 and 12 rudd at site 4). BIC varies based on ITNW. The degree of IIS is calculated as the IINW/ITNW ratio and includes values from 0 to 1. A high value means that all individuals utilize the entire niche of the species, whereas low values signify low intraspecific overlap and thus a high degree of IIS [8].

One-way analysis of variance (ANOVA) and post-hoc Tukey test were used to test the effect of site trophic state on the degree of IS. Two-way ANOVA, where the effect of tropic state was considered, was used to test whether the degree of IIS differs between the species. Data normality and homogeneity of variance were always tested prior to the ANOVA.

To determine short-term versus long-term IIS, *δ*^13^C and *δ*^15^N of body tissues with different isotopic half-lives were compared: plasma 20 ± 4 days, blood 33 ± 6, fin 36 ± 6, muscle 76 ± 28 (See Appendix A for the calculation and mean isotopic half-lives for all tissues considering the mean size of the species). Six pairwise comparisons for each individual species were performed: blood × fin, plasma × blood, plasma × fin, blood × muscle, fin × muscle, plasma × muscle (sorted in ascending order based on the half-life difference between the two tissues). The variation in isotopic signal of tissues from one individual represents IINW, and the interindividual variation represents IBIC. IINW, IBIC, ITNW, and degree of IIS were calculated from *δ*^13^C in the Ind Spec1 program [47].

A linear mixed effect model with second order polynomial and site identity used as a random effect was applied to test the relationship between range of IINW and individual size for each fish species. The function *lmer* in package lme4 version 1.1–27.1 was used [48]. Linear regression was used to test the relationship between the IS and trophic state for each site and each species. Statistical analysis was conducted using R software version 4.1.1 exclusively [49].

## 3. Results

### 3.1. TNW in Relation to Site Trophic State

European catfish occupied the largest ITNW represented by standard ellipse areas (SEAcs). The mean SEAc of European catfish was 0.179, i.e., it was 1.9 to 2.6 times larger than that of the other species (Figure 3; Table 2). In contrast, Northern pike had the smallest average ITNW, followed by Eurasian perch and rudd. Variability in ITNW among sites was highest for the European catfish, followed by rudd, whereas it was relatively low among populations of Northern pike and Eurasian perch. No clear relationship was found between ITNW and site trophic state. The greatest ITNW, along with the greatest variability among species, was observed at sites 2 and 3 with a mean SEAc of 0.127 (± 0.041) and 0.125 (± 0.1), respectively (Table 2). The second smallest SEAc of the populations at site 1 reached 0.095 (± 0.014). The smallest ITNW and lowest variability were observed on average at site 4 with the lowest trophic state, where the mean SEAc was 0.0855 (± 0.058).

### 3.2. IINW in Relation to Body Mass

A clear trend in IINW was observed for all species as a function of body mass. A decreasing IINW with body mass was observed for European catfish, Northern pike, and Eurasian perch at all sites. However, when the body mass of European catfish and Northern pike reached a threshold value, a significant increase in IINW was recorded. The thresholds for European catfish and Northern pike seem to be 11 and 5 kg, respectively (Figure 4a,b). A second-degree polynomial best fitted this trend. For European catfish: linear term (W χ^2^ = 19.97, df = 1, *p* < 0.001) and quadratic term (W^2^ χ^2^ = 26.83, df = 1 *p* < 0.001). For Northern pike: linear term (W^2^ χ^2^ = 32.42, df = 1, *p* < 0.001) and quadratic term (W^2^ χ^2^ = 52.11, df = 1, *p* < 0.001). For Eurasian perch (mesopredator), this trend was also clearly present but could not be conclusively proven, likely due to the small number of individuals growing to more than 1 kg of body mass. However, a slight increase in IINW was observed in the largest individuals (Figure 4c). Values for Eurasian perch did not show an ideal polynomial trend, but were still statistically significant: linear term (W χ^2^ = 13.43, df = 1, *p* < 0.001) and quadratic term: (W^2^ χ^2^ = 11.57, df = 1, *p* < 0.001). However, when the three largest individuals were excluded, the trend became a linear decrease in IINW with body mass (W^2^ χ^2^ = 21.51, df = 1, *p* < 0.001) (Figure 4c). In contrast to predators, an opposite trend was observed for herbivorous and generalist rudd at all sites (Figure 4d). IINW increased significantly with increasing body mass (linear term W χ^2^ = 19.97, df = 1, *p* < 0.001).

Therefore, our data indicate the dependence of IINW on the trophic position of individuals, IINW increases from herbivores to generalists, then decreases towards predators, but increases again when the individual reaches a threshold in body mass corresponding to the role of apex predator in the ecosystem. IINW is highest at this point. This can be expressed by a third-order polynomial (Figure 4e).

### 3.3. IIS in Relation to Species Position in the Food Web and Site Trophic State

European catfish had the lowest degree of IIS (i.e., values closer to 1), the mean degree of IIS was 0.41 ± 0.32 (Table 2). In contrast, Northern pike had the highest degree of IIS (mean degree of IIS = 0.15 ± 0.17). The mean degree of IIS for the Eurasian perch and rudd populations were 0.23 (± 0.18) and 0.2 (± 0.22), respectively (Figure 5, Table 2). This difference in the degree of IIS between species was statistically significant (F1,15 = 9.131, *p* < 0.01).

The degree of IIS differed significantly between oligotrophic and mesotrophic sites for both European catfish and Northern pike (Figure 5, Table 3). The degree of IIS decreased significantly with an increasing site trophic state. IIS was 6 and 3.7 times higher for Northern pike and European catfish at oligotrophic sites 1 and 2, respectively, than at mesotrophic sites 3 and 4 (Figure 5, Table 3). No relationship between the degree of IIS and site trophic state was observed for the Eurasian perch and rudd (Figure 5, Table 3). The IIS of perch at oligotrophic sites was, on average, only 1.6 times higher than at mesotrophic sites, so the site trophic state does not seem to be the decisive factor influencing IIS. For rudd, the IIS was 1.1 times higher at mesotrophic sites than at oligotrophic sites, but variability among sites with similar trophic states was even greater (Figure 5, Table 3).

A significant decrease in IIS with increasing site trophic state was observed for the mean degree of IS of all species combined (F1,18 = 6.309, *p* < 0.05). The mean degree of IIS of all species at sites 1, 2, 3, and 4 were 0.1, 0.2, 0.27, and 0.42, respectively.

In addition, the duration of IIS was determined to reflect the seasonal food supply. As mentioned above, European catfish and Eurasian perch exhibited a relatively low degree of IIS that decreased over time, based on the variance in isotopic half-lives in two different tissues. However, the dependence of IIS on variance in half-lives was low in both European catfish (R^2^ = 0.117) and Eurasian perch (R^2^ = 0.128) (Figure 6). Northern pike and rudd, species with a relatively high degree of IIS, showed no trend towards decreasing IIS over time (Northern pike R^2^ = 0.003, rudd R^2^ = 0.004; Figure 6).

## 4. Discussions

The trophic state of the environment, and thus the availability of food, is generally considered a key factor influencing foraging behavior [16]. Surprisingly, however, it does not appear to be the key factor affecting species ITNW, the most general aspect of foraging behavior. Although the smallest average ITNW was observed in the reservoir with the highest trophic state, there was considerable variability among species ITNW sizes and among sites. The combination of environmental heterogeneity, species richness, intraspecific or interspecific competition, and predation pressure will influence ITNW more than the site trophic state itself [50].

As we hypothesized, the largest ITNW was achieved by the apex predator, the European catfish. This can be explained by the theory of generalism of large-bodied apex predators, with their high energy requirements and minimum foraging constraints they utilize a larger percentage of the ITNW [9]. In contrast, the ITNW of the other species were relatively similar to each other. The Northern pike had the smallest ITNW, being an obligate predator feeding exclusively on moving prey [51]. The trophic position of Northern pike at all study sites is just below that of European catfish, which appears to control the species through both competition and predation [52,53]. A significant influence of European catfish was evidenced by the finding that the smallest ITNW of all species and at all sites for Northern pike was observed at site 4, where the catfish biomass was highest [34]. In contrast, the ITNW of Northern pike was greatest at sites 2 and 3, where the European catfish biomass was lowest [34]. Thus, considering the ecological factors, the results prove that the Northern pike is a foraging specialist [12]. The ITNW size of the other species (Eurasian perch, rudd) was between the European catfish and the Northern pike. Eurasian perch has the role of a mesopredator with a wide range of prey (fish, zoobenthos, zooplankton) [22]. Similarly, rudd can feed on a relatively wide range of food sources, from various species of macrophytes to zoobenthos [36], and at larger sizes can rarely switch to predation [10]. Due to sufficient food availability, grazers are not pushed to become foraging specialists, in contrast to obligate predators, which are expected to have high requirements for learning specific hunting skills [14]. Based on the above, our first hypothesis (ITNW of a species and a population will increase with both trophic position in the food web and with the site trophic state) was not consistent with our observation and the process is significantly more complex and depends also on other factors.

The most surprising results emerged from the IINW in relation to fish body mass. We assumed decreasing IINW with increasing body mass. This strategy minimizes the time required to search for and handle prey and is consistent with optimal foraging theory [14], but it could not be clearly demonstrated across the entire size spectrum of any of the species. The trend was clear in smaller individuals of apex predators, obligate predators, and mesopredators. All individuals were adults, but with significantly different body masses, typical of ectotherms with indeterminate body growth [54]. European catfish and Northern pike clearly exhibited the Apex predator effect, in which feeding behavior aimed at increasing specialization suddenly changes to generalism after reaching a certain threshold of body mass. This effect was particularly evident when all individuals from all study sites were considered. The thresholds of body mass of European catfish and Northern pike were about 10 kg and 5 kg, respectively. The effect was also partially observed for the mesopredator (Eurasian perch), but only three very large individuals were caught (2 × 0.9 kg; 1 × 1.9 kg). IINW size seemed to increase rapidly for these largest individuals. However, due to the limited data set, the result was not statistically significant. Future studies could focus on the IINW of Eurasian perch or other smaller predators at sites where they are the largest individuals or where large-bodied apex predators are not present, so that perch has the role of the apex predator [52], and the peak predator effect can be observed. Therefore, the findings about IINW correspond with our second hypothesis (IINW decreases with a body mass increase). However, in addition, the apex predator effect was observed when IINW increases in the largest individuals.

Thus, generalism above a certain threshold of body mass may be determined by the attainment of the highest trophic position. As an apex predator, a species has no limitation in its foraging behavior, has high nutritional requirements [9], and does not need to minimize foraging time thanks to the low predation risk [14]. The second, less likely reason could be the fact that specialists tend to be more vulnerable to environmental changes than generalists, which are able to adapt better [55]. This theory can also be applied at the individual level. Older and larger individuals are expected to have repeatedly encountered critical periods of food shortage, and this ‘die or adapt’ selection pressure [56] forces them to expand their food spectrum. Our results bring forth a fundamental question regarding the differences in the ways ectothermic and endothermic species rise to the position of apex predator. Endothermic vertebrates, mammals and birds, are already born as apex predators [53] thanks to parental care and common group formation [57], whereas ectothermic vertebrates with indeterminate growth must reach the apex predator position via lower trophic positions [58]. Only some species, and only a limited number of individuals, can reach this position [8,59]. Thus, for an ectothermic species, the apex predator status is not directly linked to the species but rather to the size of the individuals.

The size of IINW of the herbivorous generalist rudd follows the opposite trend from what we expected. The IINW size increased with body mass, but the reason is not entirely clear. Ectotherms with a tendency to herbivory often switch from omnivory to herbivory over time [60], so the IINW size should decrease. Rudd have a pronounced seasonal variability of diet depending on temperature [11]. Sampling was carried out at the end of summer, so SIA should mainly reflect the growing season with abundant plant food and an ideal temperature for cellulose digestion (See Appendix A for the isotopic half-lives of rudd tissues), rather than the cold period when the rudd diet changes significantly [11]. A possible explanation could be the ability of large rudd individuals to digest a more diverse plant diet (for instance, with low levels of toxic substances) that smaller individuals cannot digest due to the physiological limitations [60], or the occasional piscivory diet [10].

The four described trends of IINW were combined in the increasing order of the fish trophic position. This theoretical IINW model for the entire food web suggests that the IINW of herbivores is small, it increases in generalists, decreases in predators, and is greatest in apex predators. It would be prudent to corroborate these results with a similar study design using a similar trophic position of different fish species in a different ecosystem, or even with endothermic vertebrate species. The latter model could include more than four species in the food web with fewer individuals per species.

When multiple species coexist in an ecosystem, their partitioning of food resources is usually specialized [21]. According to our observations, unlike ITNW, the extent of IIS is significantly influenced by the site trophic state. This is particularly evident in apex and obligate predators and is consistent with the theory that individuals are forced to become more specialized when resources are limited [16,25]. The trend is more evident at the apex predator position, likely due to an overall 90% energy loss at each level in the food web [61]. Therefore, predators, especially apex predators, face greater resource limitations than species at lower trophic positions and must respond more strongly to site trophic decline. In general, exposure to more demanding conditions leads to greater inductive specialization [25].

Contrary to the third hypothesis, the degree of IIS did not increase with trophic position [62]. This trend was partially observed only at oligotrophic sites 1 and 2. In contrast, the trend was slightly opposite at mesotrophic sites 3 and 4. There, the IIS of the apex predator was extremely low. Thus, the overall trend of IIS apparently changes at the position of the apex predator, similar to the IINW closely associated with IIS [25,63]. Although these results are intriguing, we are well aware of the limitations of the selected study sites, where two oligotrophic and two mesotrophic sites have very similar TP concentrations. Accordingly, it is necessary to approach these findings with caution. The present study shows the direction; however, further research will be needed into this important, yet so far largely neglected, phenomenon we have highlighted above.

The degree of IIS in relation to tissues differing in isotopic half-life indicates whether the IIS is short-term (seasonal) or long-term in nature. The degree of IIS of European catfish (apex predator) and Eurasian perch (mesopredator) was generally low and even lower in the long term. Switching between food sources is thus very common during the season depending on availability, but this observation needs to be confirmed by replicating the results in more studies. This behavior is fully consistent with the Optimal Foraging Theory, which states that individuals use the most optimal and available resources. When the previously available resource becomes scarce, individuals switch to the most readily available food source [20]. Both European catfish and Eurasian perch are characterized by high food plasticity [7,9]. Low IIS with instability over time is probably a general trend for apex predators and mesopredators, consistent with their role in the ecosystem [8,52]. In contrast, Northern pike as an obligate predator generally has a high IIS. It is obviously a long-term pattern, as signals from different tissues with very different isotopic half-lives were similar. The low variability of tissues with similar isotopic half-lives was likely due to the sporadic consumption of low abundance food sources that were not present at all in tissues with a long half-life [64]. The long-term specialization has also been observed in other typical obligate predators, seabird Northern gannet (*Morus bassanus* [65]) or the loggerhead sea turtle (*Caretta caretta* [13]), both with extensive migration. Thus, it seems that food specialization at the individual level is a typical feature of obligate predators, but not of apex predators or mesopredators.

Contrary to our expectations, the isotopic signals of rudd did not show short-term IIS. Rudd commonly switch between omnivorous and herbivorous feeding depending on water temperature [11], thus, a significant difference in isotopic signals was expected. The uniform signals revealed by our data were probably caused by sampling at the end of the growing season, when all tissues already reflected plant nutrition from the warm part of the year. Variability among individuals could be caused by (i) the largest individuals, probably due to their physiological abilities, exploiting a more diverse diet (as mentioned above in the context of IINW), (ii) a probable specialization of some individuals on nutrient-rich foods (flying insects or benthos [60]), or (iii) by a difference between males and females, which are forced to use more nutrient-rich food due to gonad formation [17]. However, sex was not determined, and we plan to take this factor into account in future studies. In order to obtain the entire variability, it would be advisable to collect tissue samples several times a year, especially when the water temperature drops below 15 °C [11]. Our fourth hypothesis (IIS will exhibit short-term (seasonal) fluctuations with a decreasing trend in the degree of IIS over time) is not entirely in accordance with our observations, and it must be emphasized that the stability of IIS over time is significantly species specific. In conclusion, it is important to highlight that our study was directed towards adult (i.e., sexually mature) individuals, and the effect of gender was not accounted for due to the non-lethal sampling. However, both the developmental stage of the individual and its gender may affect the individual foraging behavior [21]. However, these specific characteristics were beyond the scope of the current research.

## 5. Conclusions

Our results show that the site trophic state has no significant impact on ITNW, but a significant impact on IIS, especially on the IIS of species in higher trophic positions. IINW is significantly influenced by the trophic position of the species and its magnitude reaches two maxima in the positions of generalist and apex predator. The findings were presented on four freshwater fish, but they are likely to possess broader applicability. In order to validate and strengthen these findings, it is recommended to replicate similar studies in different aquatic as well as in terrestrial environments.

## Figures and Tables

**Figure 1 biology-12-01113-f001:**
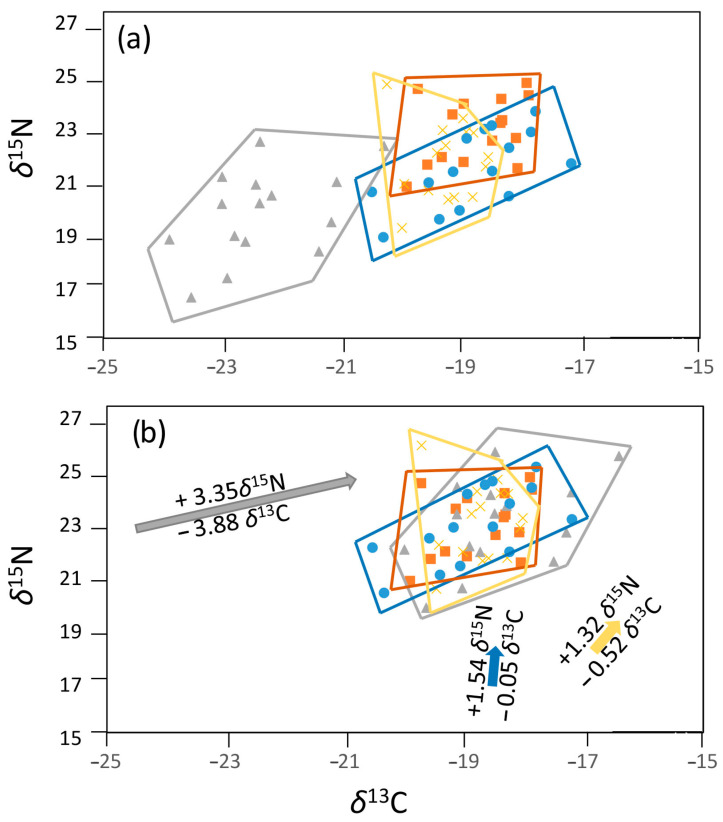
Centering process on the example of fifteen rudd individuals from the Milada lake. The *δ*^13^C and *δ*^15^N of the muscle (brown squares), the plasma (gray triangles), the blood (yellow crosses), and the fin (blue circles) are shown in the position (**a**) before and (**b**) after the centering. The arrows show how much the tissues were shifted by the centering process, the direction, and length of the shifts. The shifted *δ*^13^C and *δ*^15^N values, representing the discrimination factor to the muscle tissue, are based on the difference in the mean value of *δ*^13^C and *δ*^15^N between the muscle and each tissue. The mean values were determined separately for each tissue by averaging 46.7% values from seven individuals whose values represented the two central quartiles. Four individuals with the lowest and the highest values of both *δ*^13^C and *δ*^15^N were not included as their values could be significantly affected by different diets in a certain period. The same process was conducted for all four species and all four study sites, as each species and partly each study site showed different discrimination factors for each tissue.

**Figure 2 biology-12-01113-f002:**
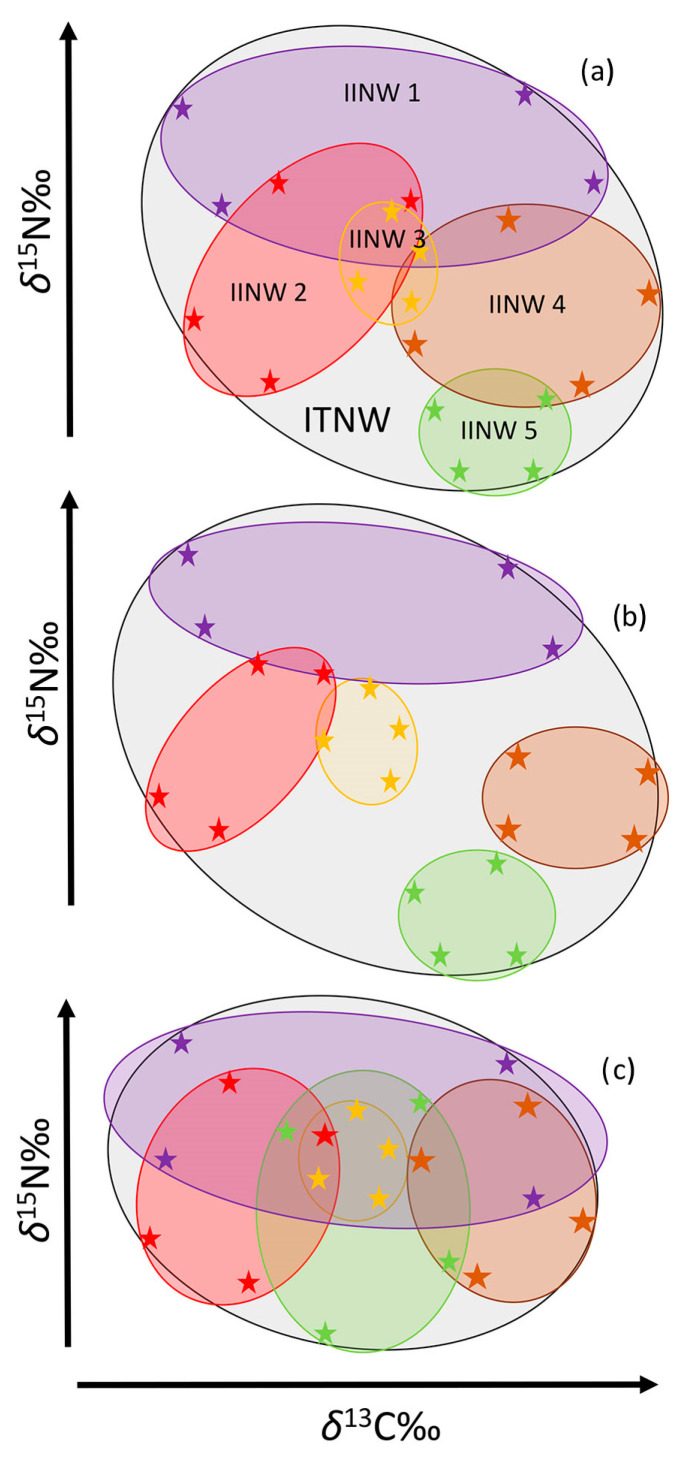
Models illustrating diet niche characteristics according to SIA. (**a**) The black ellipse represents isotopic total niche width (ITNW) expressed by SEAc (sample-size corrected standard ellipse area). Each ITNW contains isotopic individual niche widths (IINWs). Only five of them in different colors are shown for clarity (however, 15 ind. of each species at each study site were used). Each IINW is formed by four points representing the signals from four body tissues with different isotopic half-lives (plasma, blood, fin, muscle). Areas of ITNW, where IINWs do not overlap, represent the IBIC (isotopic between individual component of variation). (**b**) Example of a population with a high IBIC value and with high individual specialization. (**c**) Example of a population with a low BIC value and low individual specialization.

**Figure 3 biology-12-01113-f003:**
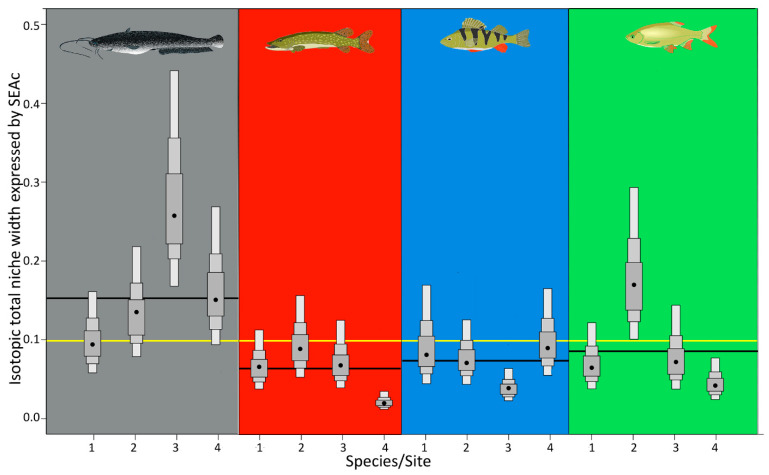
Isotopic total niche width (ITNW) expressed by SEAc (sample-size corrected standard ellipse area) for European catfish (gray), Northern pike (red), Eurasian perch (blue), and rudd (green) at sites 1–4 (sorted by increasing site trophic state). Boxes show the 95, 75, and 50% Bayesian credibility intervals for estimates based on the SIBER model. The black horizontal lines show the mean ITNW of species included at the sites. The yellow line shows the mean ITNW of all species at all sites.

**Figure 4 biology-12-01113-f004:**
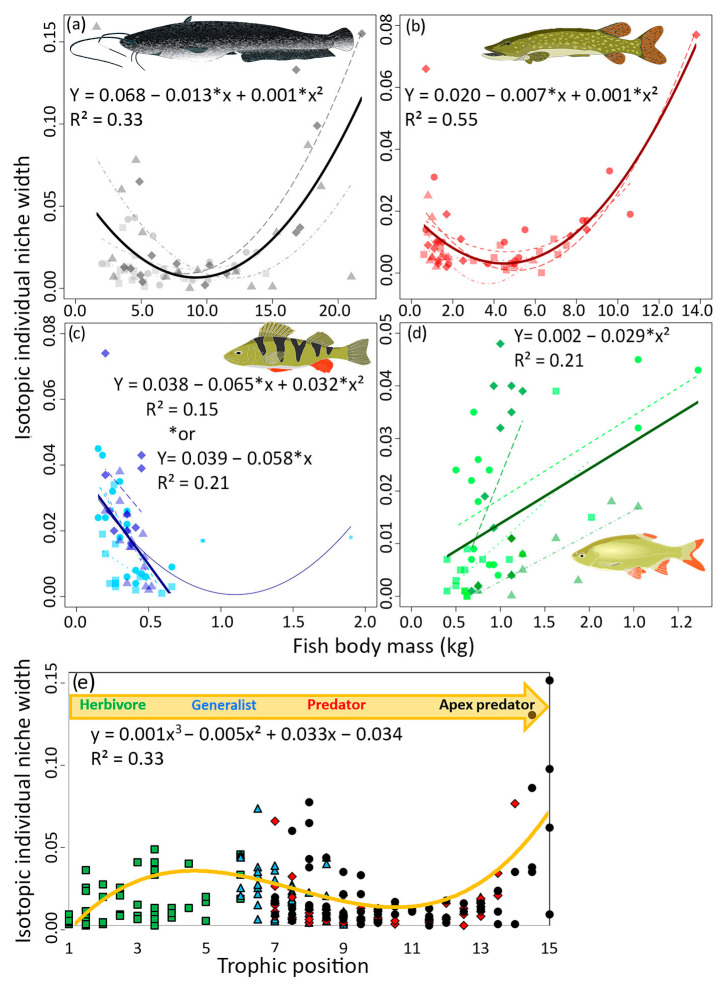
The dependence of isotopic individual niche width (IINW) on the body mass of (**a**) European catfish (black), (**b**) Northern pike (red), (**c**) Eurasian perch (blue), and (**d**) rudd (green). The dashed lines show the trends for study sites 1–4. Each study site has a different symbol. The darker the symbols, the higher the site trophic state. The solid line shows the mean trend of IINW averaged over all four sites. For Eurasian perch, we used two functions: a second-degree polynomial for all individuals and a linear regression, excluding the three largest individuals from site 1 due to the small number of large individuals reaching more than 1 kg. (**e**) Theoretical dependence of IINW on the trophic position of individuals.

**Figure 5 biology-12-01113-f005:**
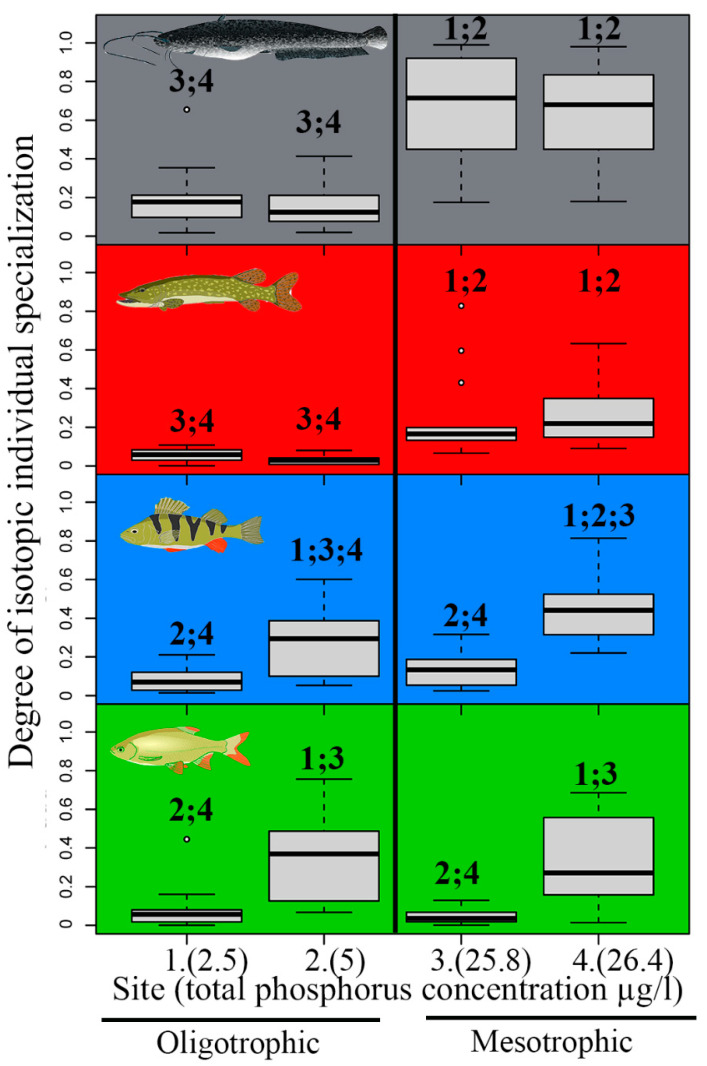
Degree of isotopic individual specialization of European catfish (gray), Northern pike (red), Eurasian perch (blue), and rudd (green) at two oligotrophic and two mesotrophic study sites (total phosphorus concentration stated in brackets). The lower values means higher isotopic individual specialization. Box and whisker plots: upper and lower quartiles (boxes), median values (line inside the boxes), maximum and minimum values (whiskers), and outliers (circles) are shown. Numbers above the box and whisker plots present the study sites, where the difference was statistically significant (1: Most lake, 2: Milada lake, 3: Žlutice reservoir, 4: Římov reservoir). See Table 3 for the statistical results.

**Figure 6 biology-12-01113-f006:**
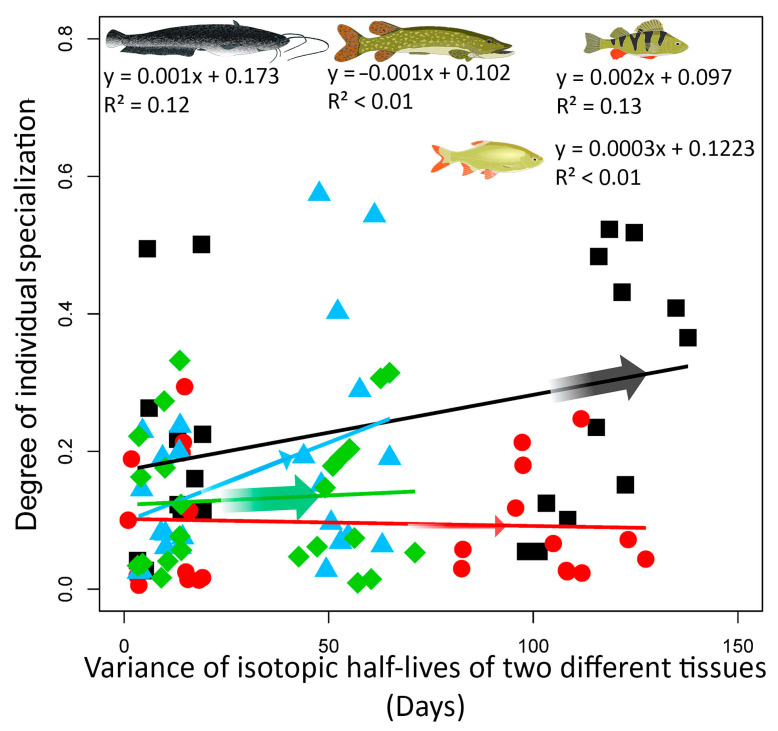
The graph shows European catfish (black), Northern pike (red), Eurasian perch (blue), and rudd (green) from all sites as a function of variance in isotopic half-lives in two different tissues (a pairwise comparison of plasma, fin, blood, and muscle).

**Table 1 biology-12-01113-t001:** Basic hydrological and geographical parameters of the four study sites.

Site	Area (ha)	Volume (10^6^ m^3^)	Long-Term Flow (m^3^/s)	Detention Time (Days)	Max. Depth	Altitude (m a.s.l)	Trophic Status, TP (μg L^−1^)
1. Most	311	70	0.06	Drainless	75	199	2.5
2. Milada	250	36	0.04	10,248	25	145	5
3. Žlutice	161	16	1.24	146	23	508	25.8
4. Římov	210	34	4.38	93	45	468	26.4

**Table 2 biology-12-01113-t002:** Summary of min, max, and mean ± SD of fish body mass, total niche width (ITNW), individual niche width (IINW), and individual specialization (IIS) of four fish species at the four study sites (1 = Most, 2 = Milada, 3 = Žlutice, 4 = Římov). N means the number of studied individuals. When applicable, ±SD is present.

Species	Site	N	Mass (kg)	ITNW (SEAc)	IINW (SEAc)	IIS
Catfish	1	16	2.4–14.5 (5.1 ± 2.78)	0.105	0.011 (±0.008)	0.1902 (±0.151)
	2	16	2–13.8 (7.87 ± 3.56)	0.141	0.017 (±0.012)	0.1609 (±0.120)
	3	18	1.6–21 (9.69 ± 5.84)	0.296	0.033 (±0.041)	0.6673 (±0.283)
	4	15	3.7–21.8 (10.59 ± 5.90)	0.173	0.041 (±0.048)	0.6310 (±0.268)
Pike	1	12	0.6–7.5 (4.7 ± 2.13)	0.105	0.006 (±0.003)	0.0554 (±0.033)
	2	14	0.7–10.6 (5.06 ± 3.09)	0.103	0.013 (±0.010)	0.0314 (±0.024)
	3	13	0.8–5.9 (1.85 ± 1.58)	0.077	0.008 (±0.007)	0.2649 (±0.227)
	4	15	0.7–13.8 (3.4 ± 3.65)	0.022	0.017 (±0.022)	0.1511 (±0.156)
Perch	1	15	0.18–1.9 (0.51 ± 0.43)	0.102	0.011 (±0.009)	0.0771 (±0.058)
	2	15	0.15–0.66 (0.33 ± 0.13)	0.078	0.021 (±0.013)	0.2767 (±0.185)
	3	16	0.2–0.52 (0.39 ± 0.09)	0.042	0.016 (±0.010)	0.1334 (±0.085)
	4	11	0.2–0.45 (0.33 ± 0.09)	0.102	0.032 (±0.016)	0.442 (±0.170)
Rudd	1	15	0.16–0.81 (0.31 ± 0.18)	0.071	0.007(±0.009)	0.0789 (±0.111)
	2	15	0.2–1.29 (0.48 ± 0.33)	0.187	0.021 (±0.013)	0.3449 (±0.232)
	3	11	0.3–1.02 (0.58 ± 0.23)	0.082	0.007 (±0.006)	0.0495 (±0.0439)
	4	12	0.27–0.5 (0.40 ± 0.07)	0.045	0.023 (±0.016)	0.3245 (±0.240)

**Table 3 biology-12-01113-t003:** Statistical comparisons of the degree of IIS of individual species at the four study sites (oligotrophic sites: 1 and 2; mesotrophic sites: 3 and 4).

Species	ANOVA	Study Site Comparisons, Post-Hoc Tukey Test					
		1 vs. 2	1 vs. 3	1 vs. 4	2 vs. 3	2 vs. 4	3 vs. 4
Catfish	(3.61) = 25.69, *p* < 0.001	*p* > 0.05	*p* < 0.001	*p* < 0.001	*p* < 0.001	*p* < 0.001	*p* > 0.05
Pike	(3.50) = 10.85, *p* < 0.001	*p* > 0.05	*p* < 0.01	*p* < 0.01	*p* < 0.001	*p* < 0.001	*p* > 0.05
Perch	(3.53) = 19.34, *p* < 0.001	*p* < 0.001	*p* > 0.05	*p* < 0.001	*p* < 0.05	*p* < 0.05	*p* < 0.001
Rudd	(3.50) = 10.12, *p* < 0.001	*p* < 0.001	*p* > 0.05	*p* < 0.01	*p* < 0.001	*p* > 0.05	*p* < 0.01

## Data Availability

The data analysed during this study are included in the Appendix A file.

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
