# Peer review of "Trophic Position of the Species and Site Trophic State Affect Diet Niche and Individual Specialization: From Apex Predator to Herbivore"

_biology, 2023, doi:10.3390/biology12081113_

Round 1
Reviewer 1 Report
1. The definition of IIS should be clarified. In line 279-280, low value means greater, but in line 354, Northern pike had the highest degree? And in line 362, the lower degree means higher individual specialization. In line 385, he "low level" is confusing.
Author Response
Response to Reviewer 1
The definition of IIS should be clarified.
In line 279-280, low value means greater, but in line 354, Northern pike had the highest degree? And in line 362, the lower degree means higher individual specialization. In line 385, he "low level" is confusing.
Thank you for the comment.
The abbreviation IIS is explained on Line 121, and the calculation of the degree of IIS is explained in the paragraph starting on Line 276.
You are right the use of different terms was confusing. We have unified the terminology. See the manuscript with Track changes.
Reviewer 2 Report
Review for the paper "Trophic position of the species and site trophic state affect diet niche and individual specialization: from apex predator to herbivore" by Lukáš Vejřík, Ivana Vejříková, Petr Blabolil, Zuzana Sajdlová, Luboš Kočvara, Tomáš Kolařík, Daniel Bartoň, Tomáš Jůza, Marek Šmejkal, Jiří Peterka, and Martin Čech submitted to "Biology".
General comment.
The isotopic compositions of animals, stable by nature, are determined predominantly by their dietary intake. The majority of studies concentrating on stable isotopes aim to discern which food sources hold the utmost importance in supporting animal consumers within food webs. The capacity of a species to fulfill its energetic necessities at a specific trophic level is dependent on several factors. These include the availability of basal resources, the efficiency of trophic transfer and the morphological as well as behavioral constraints on a predator’s aptitude for consuming available prey. Especially in aquatic ecosystems, the transmission of energy via a food web is postulated to be strongly throttled by body size, due to the restrictions associated with gape-limited predation. This is a morphological restriction, especially predominant amongst fish species. In the presented study, the authors selected four species of fish that encapsulate the entire spectrum of trophic positions: from apex predator via obligate predator to mesopredator and concluding with a predominantly herbivorious species. Employing common fish species, namely, the European catfish, Northern pike, Eurasian perch, and rudd, they aimed to disclose a number of crucial indices. These include the isotopic total niche width, isotopic individual niche width, and isotopic individual specialisation. The authors resorted to employing non-lethal methods in order to collect tissue samples. The findings reveal that the status of an apex predator is connected more closely to its size rather than the particular species it belongs to. At the lowermost level of the trophic hierarchy, body mass remains a consequential factor, albeit demonstrating a contrasting pattern when juxtaposed with apex predators. This thoroughly researched paper is well-written, complemented with germane figures and tables. The discussion focuses in on the significant findings; however, some revisions are required to fix minor concerns prior to the finale acceptance of this excellent paper.
Recommendations:
Line 19-20: We note an inconsistency wherein the statement "human-induced eutrophication can significantly change the foraging behavior" appears in the Simple Summary yet is absent in the Abstract. This omission should be amended.
Table 1: Please change the term "Trophy" to "Trophic status".
L 139-143. Provide the year for sampling periods.
Line 144: It would be beneficial if the authors could furnish further details concerning the device employed for the weighing of fish.
Table 2: Please provide the size of the sample for each species at every sampling location, possibly in a newly added footnote. Additionally, the table should include ranges, specifically the minimum and maximum levels for the body mass and indices.
The presented data do not seem to include the sex of the examined fish subjects. The authors' use of a non-lethal method excluded the need for dissection, however this prompts the question: Do the dietary habits of female and male fish differ? Relevant references in relation to this would be appreciated. Additionally, the exclusion of this data in the current research should be addressed in the limitations. Further details concerning the age and maturity status of the fish should be provided in the Materials and Methods section.
The authors decided to utilize a parametric method (ANOVA) to compare the data across the study sites. Such an approach requires the data to be normally distributed and the variances to exhibit homogeneity. Clarification is needed on whether these assumptions were tested against, requiring pertinent updates to the text.
Towards the conclusion, the research intimates that their findings are applicable to terrestrial ecosystems. However, there seems to be no substantial evidence supporting this claim within the discussion. In relation to this, the authors should interpret their results and extrapolate to terrestrial systems cautiously.
Specific remarks.
L 70 Consider replacing “recent study demonstrated” with “a recent study demonstrated
L 224. Consider replacing “different diet” with “different diets”
L 226. Consider replacing “factor for” with “factors for”
L 265. Consider replacing “between individual component” with “between individual components”
L 282. Consider replacing “One-way analysis of variance (ANOVA) and Post-hoc Tukey test was used” with “One-way analysis of variance (ANOVA) and Post-hoc Tukey test were used”
L 339. Consider replacing “Therefore, our data indicates” with “Therefore, our data indicate”
L 377. Consider replacing “with similar trophic state” with “with similar trophic states”
L 446. Consider replacing “the findings about IINW corresponds” with “the findings about IINW correspond”
L 489. Consider replacing “face greater resource limitation” with “face greater resource limitations”
Minor
Author Response
Response to Reviewer 2
Line 19-20: We note an inconsistency wherein the statement "human-induced eutrophication can significantly change the foraging behavior" appears in the Simple Summary yet is absent in the Abstract. This omission should be amended.
Thank you for the comment. We have unified the information about eutrophication in both Simple Summary and Abstract.
Table 1: Please change the term "Trophy" to "Trophic status".
We have changed the term.
L 139-143. Provide the year for sampling periods.
We have added the year of sampling, see L 142.
Line 144: It would be beneficial if the authors could furnish further details concerning the device employed for the weighing of fish.
We have added the information about the weighing of fish. L 151–153.
Table 2: Please provide the size of the sample for each species at every sampling location, possibly in a newly added footnote. Additionally, the table should include ranges, specifically the minimum and maximum levels for the body mass and indices.
Thank you, we have added all suggested information. See Table 2.
The presented data do not seem to include the sex of the examined fish subjects. The authors' use of a non-lethal method excluded the need for dissection, however this prompts the question: Do the dietary habits of female and male fish differ? Relevant references in relation to this would be appreciated. Additionally, the exclusion of this data in the current research should be addressed in the limitations. Further details concerning the age and maturity status of the fish should be provided in the Materials and Methods section.
Thank you for the comment. In the Methods, we mentioned that we focused on the adult fish (mature individuals). Lines 142–143.
At the very end of the Discussion, we mentioned that individual foraging behavior can be influenced by the gender as well as the developmental stage of the individual. However, the study was focused only on the adult life stage and due to non-lethal sampling, the effect of gender was not included. The non-lethal sampling is more important for the study than the potential effect of the gender. For the statement, see Lines 551–555.
The authors decided to utilize a parametric method (ANOVA) to compare the data across the study sites. Such an approach requires the data to be normally distributed and the variances to exhibit homogeneity. Clarification is needed on whether these assumptions were tested against, requiring pertinent updates to the text.
Yes, data normality and variance homogeneity were always tested. Thank you for the comment; we have added the information to the text. Lines 294–295.
Towards the conclusion, the research intimates that their findings are applicable to terrestrial ecosystems. However, there seems to be no substantial evidence supporting this claim within the discussion. In relation to this, the authors should interpret their results and extrapolate to terrestrial systems cautiously.
Thank you for the comment. In the Simple summary, Abstract and also Conclusion, we have moderated the claims about the general validity of our findings and recommended conducting similar studies in different environments to strengthen our findings.
Specific remarks.
L 70 Consider replacing “recent study demonstrated” with “a recent study demonstrated
Corrected
L 224. Consider replacing “different diet” with “different diets”
Corrected
L 226. Consider replacing “factor for” with “factors for”
Corrected
L 265. Consider replacing “between individual component” with “between individual components”
Thank you for the suggestion, however the term “between individual component” and the abbreviation “BIC” has been already published in other studies. Thus, we used the same term.
L 282. Consider replacing “One-way analysis of variance (ANOVA) and Post-hoc Tukey test was used” with “One-way analysis of variance (ANOVA) and Post-hoc Tukey test were used”
Corrected.
L 339. Consider replacing “Therefore, our data indicates” with “Therefore, our data indicate”
Corrected.
L 377. Consider replacing “with similar trophic state” with “with similar trophic states”
Corrected.
L 446. Consider replacing “the findings about IINW corresponds” with “the findings about IINW correspond”
Corrected.
L 489. Consider replacing “face greater resource limitation” with “face greater resource limitations”
Corrected.
Thank you for all comments. See the manuscript with tracked changes where you can see all the changes mentioned above.